# Exploring Factors Affecting the Adoption of IoT in Healthcare: A Systematic Literature Review

**DOI:** 10.3390/healthcare13233157

**Published:** 2025-12-03

**Authors:** Ruba Alnajim, Ali Alkhalifah

**Affiliations:** Department of Information Technology, College of Computer, Qassim University, Buraydah 51452, Saudi Arabia

**Keywords:** healthcare, IoT, Internet of Things, adoption, factors

## Abstract

**Background:** The integration of the Internet of Things (IoT) into healthcare promises significant advancements in patient care and operational efficiency, while simultaneously posing substantial cybersecurity threats to sensitive personal data. **Methods:** This paper employs a systematic literature review (SLR) to investigate the factors influencing IoT adoption in healthcare, clarify research trends, and examine the impact of cybersecurity. Eligibility criteria were established, focusing on peer-reviewed studies published between 2015 and 2025 across five reputable databases, ultimately identifying 79 relevant articles. **Results:** Analysis revealed active research in countries such as India, Saudi Arabia, the USA, China, Malaysia, and Pakistan, with notable publication peaks in 2019 and 2022. Most studies (60.8%) employed quantitative methods, reflecting a preference for statistical analysis, while personal IoT health devices—particularly wearables and smart home technologies—were prominent, referenced in 27.8% of the literature. The technology acceptance model (TAM) and its variants emerged as dominant frameworks, underscoring a focus on user acceptance. A total of 139 indicators influencing IoT adoption were categorized into individual, technological, security, environmental, and ‘other’ factors. Despite the rigorous search process, some relevant studies may have been overlooked, indicating a limitation in the review. **Conclusions:** Addressing these gaps is vital for enhancing security measures in IoT applications, contributing to data protection and ensuring continuity of care in an increasingly interconnected healthcare ecosystem. This synthesis of current research underscores the necessity for ongoing exploration of IoT’s implications for both patient care and cybersecurity in healthcare settings.

## 1. Introduction

Advancements in information technology (IT) continue to enhance service delivery across various sectors. One of the most critical sectors is healthcare which is both vital and continually evolving. The sector’s critical nature can also be attributed to the expansion of precision medicine and the urgent necessity for individuals to be aware of their medical problems in real time [1]. These advancements are resulting in today’s smart objects that are being introduced in an era of apps and devices based on the Internet of Things (IoT). The concept of integrating sensors and intelligence into objects was discussed during the 1980s and 1990s [2]. In 1999, British technology pioneer Kevin Ashton coined the term “Internet of Things” (IoT), which he developed while working at the Massachusetts Institute of Technology (MIT) [3]. Even though the phrase ‘Internet of Things (IoT)’ is extensively used, its meaning remains unclear [4]. In a study by Chebudie et al., it is defined as “a network that connects uniquely identifiable ‘Things’ to the Internet” [5]. The IoT has many uses globally, including in the environmental, transportation, agriculture, industry, infrastructure, and healthcare sectors [6]. With the IoT’s ability to facilitate connectivity between everyday objects and the internet, in healthcare, this can allow monitoring and tracking of patients’ health metrics [7], as well as adopting and integrating smart technology in healthcare, ranging from vital signs devices to managerial procedures. Furthermore, the IoT market is projected to reach USD 94.2 billion by 2026, with a compound annual growth rate (CAGR) of 28.9%, surpassing the moderate growth rate of the medical device market of 5.5% between 2022 and 2026 [8]. Nevertheless, a major concern is gaining an understanding of how individuals perceive IoT products and applications and how to facilitate the adoption of these products and applications [2].

A crucial element to encourage adoption is to explore how market dynamics and regulatory frameworks influence both innovation in medical equipment and the readiness of companies to adopt IoT technologies. Recognizing these interconnections can elucidate how collaborative actions within innovation networks improve the effectiveness and efficiency of IoT applications in enhancing patient outcomes [9].

Another important effect is the involvement of multi-doctors which positively impacts patient adherence to medical instructions. Leader participation and disciplinary variety enhance the positive correlation between multi-doctor involvement and patient adoption [10].

Finally, an important aspect of supporting the adoption of any new technology is security. Security in the context of healthcare is crucial, as the sector is increasingly pushed to adopt advanced technologies through IoT devices. The sensitive nature of patient data necessitates extreme precautions to maintain its confidentiality. Moreover, security in IoT devices is seen as a significant concern as data may be readily compromised by attackers or hackers [11]. The complexity of healthcare systems, characterized by numerous data sources and endpoints, poses distinct obstacles to the implementation of efficient cybersecurity measures [12]. Primary points of focus comprise the safeguarding of electronic health records, assurance of data integrity and confidentiality, and protection of patient privacy. Many techniques have been employed to enhance the security of IoT devices, for example, the radio-frequency identification (RFID) encryption technique [13] and a biometric-based security system [14]. Machine learning (ML) and public security-focused datasets are also being investigated to improve healthcare cybersecurity [15]. In the privacy context, healthcare deals primarily with personal data which governs its management and storage approaches using privacy-preserving techniques. The United States (USA) and the European Union (EU) have established the *Healthcare Insurance Portability and Accountability Act 1996* (HIPAA) [16] and the General Data Protection Regulation (GDPR) [17], respectively.

Furthermore, healthcare has an additional feature that increases the challenge of exceeding the current demands for privacy, namely, patient safety [15]. This issue is becoming significant owing to specific attacks on implantable and wearable medical devices (IWMDs) (insulin pumps, health trackers, etc.), which can lead to unfavorable effects. Despite these considerable efforts, security and privacy concerns in the Internet of Things (IoT) remain inadequately addressed.

To better justify the necessity of this review at this time, recent cybersecurity incidents reveal how the COVID-19 pandemic significantly expedited the growth of IoT and elevated cybersecurity risks in healthcare, with an enormous spike in hacks during this time. This study [18] concludes that cyberattacks on healthcare sectors increased by two to three times during the initial wave of the pandemic compared to 2019. Moreover, ref. [19] saw an “exponential growth of Internet of Things (IoT) devices” that significantly expanded the risk landscape for possible attacks in the healthcare sector. Consequently, this paper aims to conduct a systematic literature review (SLR) to compile and analyze the factors influencing the adoption of IoT in the healthcare sector. However, this study expands on these by introducing research gaps in three aspects, thereby contributing to the enrichment of both medical and information technology fields.

This research paper is structured as follows.

Section 2 presents the literature on the previous empirical studies. Section 3 explains the research methodology. Section 4 discusses the results obtained by categorizing the findings into six categories. Section 5 details the gaps and future agenda. Finally, the contributions and conclusions are outlined in Section 6 and Section 7.

## 2. Literature Review

### Previous Empirical Studies

Several studies have been conducted to discover the factors that affect IoT adoption in the healthcare sector. Our study’s systematic literature review (SLR) has narrowed down these studies to the five top-rated articles based on their high index score and being published in well-known journals.

One of the earliest studies, performed by [20], covered 146 articles and identified five approaches: sensor-based, resource-based, communication-based, application-based, and security-based. Each approach was evaluated and compared. In a later study, the researchers in [21] analyzed more than 106 papers, categorizing the challenges of IoT adoption into three classes: system, users, and other challenges. Similarly, the authors in [22] identified 90 factors from 22 articles categorized into five main areas. These areas were individual (16 factors); technology (19 factors); security (7 factors); health (9 factors); and environment (7 factors); this total of 90 factors included a subset of factors for each area that could be either barriers to, or facilitators of, this technology. In the fourth paper [23], the authors conducted a meta-analysis, resulting in the IoT applications in healthcare being categorized into monitoring, locating, and personalized medicine systems, each with many studies and inventions over the years. Finally, an SLR based on 10 articles was performed by [8] to discuss various theories on technology adoption, focusing on socio-technical theory. The study found 94 factors influencing IoT adoption in healthcare that could be classified into 24 themes. The results and limitations of these past studies on IoT adoption in healthcare are presented in Table 1.

## 3. Methodology

This study took a systematic literature review (SLR) approach, following the Okoli and Schabram protocol [24], with four phases: planning, selection, extraction, and execution. The SLR, focused on IoT adoption in the healthcare sector, was employed to identify, evaluate, and interpret all the literature applicable to the research questions [25]. Systematic literature reviews (SLRs) are essential for implementing evidence-based techniques across many fields, including IT, healthcare, and social sciences. They generally seek to address multiple goals: establishing a theoretical foundation for future study; obtaining knowledge of the extensive research on a subject of interest; or addressing practical inquiries by gaining an understanding of the insights provided by current research on the issue [24]. Their main advantage is that, instead of serving as a foundation for the researcher’s personal goal, they establish a robust starting point for all academic community scholars engaged in a certain field. Systematic literature reviews (SLRs) are especially helpful for meta-analytic studies that collect and synthesize information from prior investigations [26]. The SLR approach provides opportunities to minimize bias and increase clarity via a framework of guidelines and validation processes [27]. This study adheres to the PRISMA (Preferred Reporting Items for Systematic Reviews and Meta-Analyses) guidelines, ensuring transparency and completeness in reporting the methodology and findings of our systematic review. This study was registered on PROSPERO (CRD420251234879) on 19 November 2025.

### 3.1. Planning Phase

The planning phase involved the identification of the study questions and objectives. These were developed to enhance the research: recognizing and classifying the existing research literature from several sources to investigate factors and theories, either quantitatively or qualitatively, in the adoption of IoT within the healthcare context. Continually, the first phase was adapted to identify the relevant literature, guided by the following objectives:-To explore the factors affecting the implementation of IoT;-To clarify academic research trends in the field of IoT in healthcare;-To understand the role of cybersecurity and its impact on IoT within healthcare.

In light of that, the following research questions (RQs) were formulated:

Q1: What are the key factors that influence the adoption of IoT technologies in healthcare settings?

Q2: What are the main frameworks, models used?

Q3: What role do cybersecurity concerns play in the adoption of IoT technologies in healthcare?

Moreover, for this systematic literature review, papers were gathered from the following five digital databases: Emerald, ScienceDirect, Institute of Electrical and Electronics Engineers (IEEE), Springer, and Taylor & Francis, focusing on articles on IoT acceptance among healthcare workers and patients.

Furthermore, studies were chosen from the databases utilizing the given keyword combinations string: (healthcare <OR> e-Health) [AND] (IoT <OR> “Internet of Things”) (adoption <OR> implementation) (“factors”).

### 3.2. Selection Phase

Table 2 shows the eligibility criteria employed for study selection in the systematic literature review (SLR). The inclusion criteria specify that studies must focus on factors influencing the adoption of the Internet of Things (IoT) in healthcare; be published in peer-reviewed journals with a high impact factor; be written in English; be available as a full text; and be published within the study’s time frame of 2015–2025. Conversely, the exclusion criteria stipulate the removal from the selection of books, book chapters, and newspapers; studies in languages other than English; those not available as full text; and those published before 2015. These criteria ensure that the selected studies are relevant, of high quality, and contemporary to the research question.

Kitchenham and Charters [28] emphasized the significance of quality assessment in the selection of studies, also delineating criteria for estimating the relevance and rigor of research publications. Moreover, many suggestions and measurements have been proposed in various studies to enhance the efficacy of quality assessments [29]. Consequently, in addition to the previously established inclusion and exclusion criteria, this study evaluated the quality of individual articles using additional criteria. Selections, especially for conference papers, were made from well-respected conferences that often utilized a rigorous peer-review procedure to ensure elevated quality standards.

### 3.3. Extraction Phase (PRISMA Flow Diagram)

A PRISMA [preferred reporting items for systematic reviews and meta-analyses] flow diagram was used to illustrate the systematic process of the study selection for the review. Initially, 3002 records were identified through database searches. After removing duplicates (n = 2033) and records with restricted access (n = 254), 715 records underwent screening. Following the screening phase, 400 records were excluded due to being in a non-English language, leaving 315 reports to be sought for retrieval. Of these, 110 reports could not be retrieved, and 205 reports were assessed for eligibility. Finally, a total of 79 studies met the inclusion criteria and were included in the review. The exclusion criteria at the eligibility assessment stage included reasons based on the title (n = 26), abstract (n = 64), and the journal’s index (n = 36). Figure 1 presents the flow diagram [30] which provides a transparent and reproducible overview of the study selection process.

### 3.4. Execution Phase (Results)

In the execution phase, Excel spreadsheets were utilized to filter the results by summarizing the data in a form that described the researchers’ key findings. The summarized data on the form were further analyzed and translated into charts to illustrate research ratios based on the theories being used, countries where studies took place, the year of publication, research design, technologies used, and, finally, the factors influencing IoT adoption in health sectors. We sought data on factors related to IoT adoption in healthcare. For each outcome, we aimed to collect all relevant measures reported across the studies, including both qualitative and quantitative analyses. However, we prioritized studies that were open-source and available.

## 4. Findings/Results

In this chapter, results derived from the reviewed literature will be detailed, focusing on six key attributes:

First, we will focus on countries where studies took place, with an analysis of the geographical distribution of research studies, highlighting the countries involved. Also, we recorded the year of publication to provide a chronological overview of when the studies were published, illustrating trends and developments in the field over time. Moreover, we assessed the research designs, creating a description of the various research methodologies employed in the studies and categorizing them into qualitative, quantitative, and mixed methods. We also analyzed the technologies used through an examination of the different technologies utilized in the studies, showcasing how they contributed to research outcomes. Moving on to the theories used, we performed a review of the theoretical frameworks that underpin the studies explaining how these theories informed the research. Lastly, to assess factors influencing IoT adoption in healthcare sectors, we performed an exploration of the key factors that impact the adoption of Internet of Things (IoT) technologies in healthcare

Each attribute will be accompanied by detailed diagrams and tables to provide a clear and comprehensive understanding of the findings.

### 4.1. Countries Where Studies Took Place

The healthcare sector is a dynamic domain that may captivate the interest of many researchers. Hence, the literature shows diverse countries in which research has examined their intended use of sophisticated technologies, such as the Internet of Things (IoT).

Notably, countries like India, Saudi Arabia (Kingdom of Saudi Arabia [KSA]), the United States (USA), China, Malaysia, and Pakistan have a higher number of studies, indicating active research efforts in these regions. In contrast, several other countries have fewer studies, while a significant number of studies have not mentioned the study region, possibly suggesting gaps in geographic representation. In Table 3, each row represents the number of studies from specific countries. The visualization in Figure 2 highlights the global landscape of research on IoT adoption in healthcare.

### 4.2. Year of Publication

Figure 3 illustrates the distribution of publications selected for the SLR across various years. Each segment of the pie chart represents the proportion of total publications from specific years, highlighting trends in research output over time. Notably, the years 2022 and 2019 each contributed 15% of the selected publications, while 2023 accounted for 13.8%. Other years, such as 2018 and 2017, showed slightly lower percentages, at 11.3% and 10%, respectively. The visualization in Figure 3 and Table 4 helps to identify the years with the highest research activity relevant to the factors influencing the intention to adopt IoT in the healthcare sector.

### 4.3. Research Design

Figure 4 depicts the research designs employed by the reviewed studies on IoT acceptance in the healthcare sector. The pie chart shows that most studies (60.8%) utilized quantitative research methods, indicating a preference for numerical data and statistical analysis. Qualitative methods were used in 12.7% of the studies, while mixed methods accounted for 8.9%. Additionally, 17.7% of the studies did not specify their research design. The visualization in Figure 4 highlights the dominant use of quantitative approaches in examining IoT acceptance in healthcare (Table 5).

### 4.4. Technologies Used

Table 6 presents the various technologies integrated with the Internet of Things (IoT) as discussed in the reviewed papers on measuring IoT acceptance in the healthcare sector. Each row represents the number of studies that feature specific technologies. Personal IoT health devices, particularly “wearables,” are the most commonly referenced (27.8%), indicating significant interest in this area, followed by smart homes (7.6%). Other technologies, such as mobile health services (6.3%), e-health management systems (3.8%), and care delivery devices for older adults (2.5%), also appear but with fewer mentions. Most studies (31.6%) did not specify the technology used. The visualization in Figure 5 highlights the diverse technological landscape of IoT acceptance research in healthcare, emphasizing the prominence of wearable devices.

### 4.5. Theories Used

Over the years, several theories have been utilized to investigate the influencing factors affecting IoT adoption in healthcare, with a total of 28 theories or/frameworks noted during the review. Identical or derived theories were then unified.

Table 7 presents a comprehensive overview of the various theoretical frameworks identified in the literature, along with the corresponding frequency of studies employing each theory. This quantitative representation serves to highlight the prevalence and relevance of specific theories within the field of technology acceptance. Each row in Figure 6 represents the frequency of studies employing specific theoretical frameworks. The technology acceptance model (TAM) and its variants dominate the research landscape, indicating a strong focus on understanding user acceptance. Other theories, such as the unified theory of acceptance and use of technology (UTAUT), the theory of planned behavior (TPB), innovation diffusion theory (IDT), and the health belief model (HBM), are also represented, although to a lesser extent. The visualization in Figure 6 highlights the predominant theoretical approaches in current research on IoT adoption in healthcare.

### 4.6. Factors Influencing IoT Adoption in Healthcare Sectors

Upon reviewing the current research, 139 indicators that may or may not influence IoT adoption were identified; the same or duplicate meanings were next eliminated or unified through analysis of the studies. In collecting the factors, we focused on limiting our attention to the presence, or absence, of an indicator’s impact, clarifying the main factors influencing the process of IoT adoption, ultimately explaining the lack of factors, or defective factors, in the field. The indicators were subsequently categorized by classifying and analyzing the factors that influence IoT adoption into individual, technological, security, environmental, and ‘other’ categories to fulfill the study’s objectives. We have relied on established frameworks and categorizations from the prior literature to ensure consistency and comparability. This study aimed to address knowledge gaps by focusing on cybersecurity, IoT, and the healthcare sector dynamics. A brief description of each category is provided below to clarify what they mean. For a comprehensive list of all identified factors, please refer to Appendix A.

#### 4.6.1. Individual Factors

Individual factors refer to the personal characteristics and preferences of patients and healthcare providers, for example, attitudes, with [36] highlighting the significant influence of attitudes on the adoption of IoT healthcare products. By providing a real-life example, the ability to use monitoring devices was determined, meaning that more patients were monitored with fewer checkups. When [46] developed mobile phone-based diabetes monitoring (MDM) technology, they discussed how to ensure the physician’s impact on health improvement through MDM adoption. This revealed health improvement effects, with net benefits ultimately affecting the intention to use MDM technology. In contrast, ref. [31] revealed that attitudes did not directly or indirectly impact behavioral intention. The study by [72] found that preserved health risk, represented by perceived susceptibility, was frequently a non-significant determinant in the context of the same IoT-based device.

#### 4.6.2. Technological Factors

The technological aspects of IoT devices were investigated by [63] in their study, which sought to determine the significance of perceived usefulness (PU) in influencing individuals’ intention to continuously use IoT wearable devices. They emphasized the need for a comprehensive understanding of perceived usefulness (PU) to enhance usage and development of these devices. At the same time, ref. [77] observed that perceived usefulness (PU) had an impact on trust, as well as slightly affecting individuals’ intention to use medical IoT wearable devices. Furthermore, the authors of [38] stressed that the benefits of IoT applications would only accrue positively if the adoption was by institutions as well as consumers: they declared a significant direct effect of facilitating conditions on behavioral intention. In the context of designing a smart IoT home, ref. [40] revealed that facilitating conditions had a non-significant effect.

#### 4.6.3. Security Factors

Security factors refer to the measures taken to protect sensitive health data from breaches and unauthorized access, with [71] addressing this as confidentiality concerns. These authors aimed to identify factors that created obstacles to IoT adoption in the healthcare sector. They found that the security and privacy of data were the most influential factors in identifying medical Internet of Things (MIoT) challenges, although [32] did not classify this as a core factor.

Another important factor was the presence of clear objectives and plans for MIoT systems: in [33], the authors believed that the deployment of MIoT systems could be efficiently supported by clear goals and precisely outlined strategies. In contrast, ref. [75] classified government regulation as a positive factor but found that it was not statistically significant.

Moreover, previous studies [108] revealed that perceived severity and vulnerability originating from the protection motivation theory (PMT) pose higher health risks and have positively influenced the intention of medical professionals in adopting IoT.

Furthermore, according to theories like Social Exchange Theory (SET), studies suggest that concerns over weak authentication mechanisms can lead to a perceived lack of trust in IoT devices [52]. This perception negatively impacts performance expectancy and trust.

Finally, the vulnerability of unencrypted data could raise concerns regarding data breaches and privacy breaches. This issue impacts the perceived risk factor, which, as per the Technology Acceptance Model (TAM), can substantially change consumers’ attitudes towards the technology. If users see their data as insecure, their willingness to use IoT solutions reduces [39].

#### 4.6.4. Environmental Factors

Environmental factors refer to the external conditions that IoT technology can provide, such as [79] providing evidence of the positive influence of performance expectancy (PE) on behavior towards wearable devices, from the consumer perspective. Additionally, ref. [89] illustrated that performance expectancy (PE) was a core factor, especially in healthcare professionals’ adoption of IoT and big data analytics (BDA) technologies.

On the other hand, the social influence (SI) factor tended to be a controversial point in [2], in which its significant effect on behavioral intention was demonstrated based on a techno-psychological point of view. Moreover, ref. [78] concluded that social influence (SI) did not significantly influence the intention of diabetic patients to use wearable health devices that were IoT-reliant.

#### 4.6.5. Other Factors

Factors that did not fit the above categories could fit previously defined categories, such as operational efficiency, customer satisfaction, or product quality, like cost. Cost factors, by their nature, pertained to financial implications in the study by [102], which explored technologies that analyzed behaviors associated with dementia. They revealed the necessity for a cost-effective approach to efficiently collect data, while [81] categorized preserved value as a low-impact factor.

In another context, in [40], the effects of effort expectancy (EE) on smart homes were found to be greater than those of the other evaluated factors. However, ref. [52] found that EE represented an unimportant factor.

## 5. Gaps and Future Agenda

It is essential to understand the current state of research on IoT technology applications in healthcare, as measured by systematic review and bibliometric analysis, and to address the gaps, as this allows for the optimal use and enhancement of these applications and enriches future studies.

### 5.1. IoT in the Healthcare Sector

The healthcare sector is a fertile field for creativity and technological invention, making it highly suitable for IoT technology due to its connectivity ability and unlimited applications [37]. Nevertheless, the adoption of IoT technologies in the healthcare industry is progressing at a slow rate. Many reasons can be found, but some researchers [2] attributed it to prior research being primarily focused on enterprise IoT design and usage, neglecting individual perspectives and concerns. Others [86] have noted that numerous studies have provided significant insights into the adoption of IoT-based healthcare technology by employees and patients; nevertheless, the perspectives of health technicians and professionals are rarely examined. Likewise, ref. [109] indicated that challenges, such as security, privacy, standardization, and low-power operations, need to be addressed for widespread adoption. Moreover, ref. [67] indicated that most studies focus on quantitative testing of technology acceptance models; however, qualitative exploratory studies are essential to understand the reasons for and against IoT adoption and use in the healthcare sector.

Finally, the existing research has paid little attention to moderation and mediation analyses, even though these are essential for understanding how different factors influence IoT adoption outcomes. Furthermore, many studies have not examined the relationships between these factors, limiting our understanding of how they interact and affect one another.

### 5.2. Factors

After considering the final analysis results of the literature review, it is evident that individual factors (33.1%) represent the largest segment, as shown in Figure 7, signifying that personal behaviors, preferences, and adoption rates have been the primary focus of previous studies. This indicates that user involvement and user acceptability are crucial for the effective adoption of IoT technology. Technological factors (24.5%) constitute the second most significant effect, highlighting the importance of the assessment of technological advances, including hardware and software improvements, in either promoting or obstructing IoT implementation. Environmental factors encompass 14.4%, indicating the need to evaluate the impact of external conditions, such as societal attitudes, on IoT adoption or disruption. Factors not classified into any category constitute 19.4%, signifying that additional relevant variables may influence IoT adoption or disruption, indicating a diverse range of difficulties and opportunities in the field of research. Finally, security factors represent only 8.6%, despite the need for strong cybersecurity measures to address concerns related to data protection and privacy. Details of these factors are further discussed in the next section.

As shown in Figure 8a, attitude (12.4%) is the most extensively examined individual factor, underscoring the significance of user motivation and perception in facilitating IoT adoption. Conversely, aspects such as a willingness to learn, desire, and intrinsic motivation are among the least examined, suggesting a research gap that could be explored to improve user experiences and promote broader acceptance of IoT technology in healthcare.

As shown in Figure 8b, we can conclude that the most studied technological factor—perceived ease of use (PEOU) with 19.1%—can be justified as research has generally been concerned with how intuitive interfaces, training, and support may improve PEOU, thereby promoting increased adoption and use of IoT technologies. In contrast, the least studied factor is personalization (at 0.9%), indicating less of a research focus on the influence of personalization on adoption rates. Nonetheless, customization is widely acknowledged as an essential element in improving user engagement and fulfillment, especially in healthcare, where individual patient requirements might differ widely. Addressing this research gap may provide advances in IoT systems that are more adaptive, resulting in enhanced user experiences and adoption rates.

As shown in Figure 8c, social influence (SI) (31.9%) is recognized as the most extensively researched environmental factor. The high number of studies highlights the significance of social dynamics within healthcare, with research investigating how establishing a supportive group and spreading success perspectives can enhance social impact. The least studied environmental factor is physician recommendation (1.4%), which indicates a lack of understanding of the influence of trustworthy physicians’ direct referrals on IoT adoption decisions. Promoting the influence of physician recommendations could increase the reliability and credibility of IoT solutions, ultimately resulting in increased IoT adoption rates.

Additionally, as shown in Figure 8d, we can identify perceived cost (PC) (24.4%) as the most studied uncategorized ‘other’ factor, highlighting the importance of understanding how the perception of high costs can deter IoT adoption, while demonstrating that cost effectiveness can encourage adoption. In contrast, effort (1.2%) is identified as the least studied factor in this context, indicating a gap in comprehension of the effort required to implement and use IoT technologies.

Finally, as for their interaction with technological determinants, the factors shape IoT vulnerability and resilience in healthcare in different ways. For example, inadequate digital literacy among healthcare staff could delay IoT adoption, boosting susceptibility to mistakes and security breaches. On the other hand, training and education programs enhance competence, enabling staff to better utilize and secure IoT systems, thus fostering resilience [22,110]. Furthermore, a change-resistant culture or lack of digital leadership might hinder IoT integration. However, supporting leadership and an innovative culture encourages job creation and adaptation, maximizing the benefits of robust technical solutions [8,111].

### 5.3. Cybersecurity Aspects

As mentioned earlier, security factors represent only 8.6% of the total identified factors, even though cybersecurity continues to be a critical concern in IoT-based systems. In the healthcare industry, it is imperative that a patient’s health information is easily available to authorized individuals, including physicians and nurses; however, the patient’s confidential health information must remain private.

In this research, we identified 12 security-related factors, with the relevance of their value assessed in previous research by the frequency of mentions. Upon examining Figure 9, we find the following three top-mentioned factors. Privacy (26.2%) has the highest percentage of mentions, signifying that worries about the protection of personal information are a critical element, while trust (16.7%) underscores the importance of confidence in stakeholders. Security has the same percentage of mentions (16.7%) as trust, with these equal proportions highlighting the significance of secure techniques and systems in safeguarding sensitive information.

Overall, deeply focused security-related research is scarce, despite its utmost importance in supporting the process of adopting IoT technology in the health sector.

### 5.4. Future Recommendations

Based on the findings of the study regarding IoT adoption in healthcare, we believe that several recommendations can be made. Healthcare institutions should enhance cybersecurity measures and develop patient education programs to address misconceptions about IoT technologies while investing in user-friendly devices. Regulators need to establish clear guidelines on patient privacy and cybersecurity, promote standardization in device development, support research on barriers to IoT adoption, and monitor compliance with regulations. Technology developers should focus on user-centric design, incorporate advanced privacy protections, engage with stakeholders to ensure solutions meet actual needs, and invest in continuous innovation to adapt to emerging threats. By implementing these strategies, all parties can collaborate to overcome barriers to IoT adoption, ultimately enhancing patient care and safety in the healthcare sector.

As for future research paths, we recommend further investigation into topics such as Multimedia Streaming in Software-Defined Internet of Vehicles (SD-IoV) for healthcare, like OpenFlow or Raspberry Pi. In mobile healthcare contexts (e.g., ambulances, patient transport), dependable multimedia streaming is crucial. Future research may explore the potential of SD-IoV to facilitate real-time video consultations and data sharing between automobiles and medical facilities.

## 6. Contribution

This study makes several contributions to information security research and has implications for the healthcare sector. It offers an overview of current research on IoT adoption in the healthcare sector. The study seeks to make significant contributions to research, organizations, and society by addressing a critical knowledge gap in the realms of cybersecurity and IoT dynamics within the healthcare sector. By highlighting the importance of cybersecurity in healthcare sector IoT, the research aims to raise public awareness and enhance patient safety, ultimately benefiting society in its entirety. From an organizational perspective, the study’s findings will support the development of policies and strategies, enabling healthcare providers to make informed decisions about technology adoption and risk management.

From a research perspective, one of the study’s most important contributions is the examination of multiple theories and factors that influence the adoption of IoT devices, especially in the healthcare context. Additionally, the study enhances the growing body of research investigating the current state of IoT acceptance. This study is unique as it combines several aspects into a single study. It emphasizes the importance of implementing more theories, such as the fit between individuals, tasks, and technology (FITT) frameworks, which could be more comprehensive in explaining the relationship between the studied factors. Moreover, the study draws attention to the need for more deeply focused security-related research, with the findings providing guidelines for future research focused on IoT adoption within the healthcare sector.

## 7. Conclusions

This comprehensive literature review examines existing studies that analyze the factors influencing IoT adoption in the healthcare sector. This research is unique as it reviews 79 articles, while exploring the theories being used, countries where studies took place, year of publication, and research design.

The study highlights the significant potential of IoT adoption to enhance patient care. Nonetheless, IoT adoption is impeded by obstacles, such as cybersecurity threats, patient privacy issues, and disparate opinions of IoT technology among patients and caregivers.

The study categorizes IoT adoption in healthcare into five categories (individual, technological, security, environmental, and other). It identifies prevalent areas of IoT adoption and current trends. The research synthesizes key factors influencing IoT adoption from the existing literature, organizing them into different dimensions. The research offers insights for scholars and practitioners seeking to understand IoT adoption in healthcare contexts, providing a structured framework for analysis and future research.

The results reveal that countries like India, Saudi Arabia (KSA), the USA, China, Malaysia, and Pakistan had a higher number of studies, indicating active research efforts in these regions. Notably, the years 2022 and 2019 each contributed 15% of the selected publications, while 2023 accounted for 13.8%. Other years, such as 2018 and 2017, showed slightly lower percentages at 11.3% and 10%, respectively. Most studies (60.8%) utilized quantitative research methods, indicating a preference for numerical data and statistical analysis. Personal IoT health devices, particularly “wearables” and smart homes, were the most frequently referenced (27.8%) in prior studies. In total, 28 theories/frameworks were noted during the review. The technology acceptance model (TAM) and its variants dominated the research landscape, indicating a strong focus on understanding user acceptance. Furthermore, 139 indicators that may or may not influence IoT adoption have been identified. The indicators were subsequently categorized to fulfill this study’s objectives, with the factors classified into, and analyzed in, individual, technological, security, environmental, and ‘other’ categories. This study concludes that research gaps exist in studies on IoT adoption in the healthcare sector, in terms of factors in general and in cybersecurity aspects, in particular. The data not only provides a clearer picture of the current state of IoT adoption in healthcare but also paves the way for future research in these areas. This insight encourages researchers to explore underrepresented regions, theories, and methodologies, ultimately contributing to a more balanced and comprehensive understanding of the field.

As for future recommendations based on the past literature and in-depth investigations, we believe that to enhance IoT healthcare security, actionable solutions must concentrate on establishing standardized cybersecurity frameworks. For example, using blockchain for decentralized trust, strong authentication, and secure data transmission to ensure healthcare device and network interoperability [112,113,114], cross-border data governance models, which must address the confidentiality of data and privacy issues [115,116], and AI-driven threat prediction systems, particularly those employing explainable AI (XAI), are crucial for crisis prediction and transparent decision-making in critical healthcare settings [117].

Recommended research directions can be summarized as edge computing and how it should be incorporated to facilitate low-latency, real-time threat detection [118]; blockchain-based verification that can offer immutable audit trails; decentralized identity management, particularly when integrated with edge and fog computing for scalability [119]; and, lastly, privacy-preserving federated learning that enables collaborative model training among institutions without the exchange of raw data. These recommendations can build on the paper’s findings and contribute to the field [120].

Another important aspect is legal frameworks that must be continually assessed and included in system design, with adaptive procedures to handle evolving rules such as HIPAA, GDPR, and future AI Act frameworks [115].

Lastly, the study outlines the following limitations and offers recommendations for further research. Despite conducting a thorough manual web search to identify studies, the absence of some studies is regarded as a limitation of the study. These missing studies may enhance the study’s outcomes; hence, future research should adopt an automated search approach to maximize the collection of relevant studies. The inclusion and exclusion criteria chosen by the study may also be a limitation (e.g., including only full-text articles while excluding native paid content). Hence, addressing these concerns in future studies may be extremely valuable. Finally, the previously revealed gaps are regarded as an important path for further research.

## Figures and Tables

**Figure 1 healthcare-13-03157-f001:**
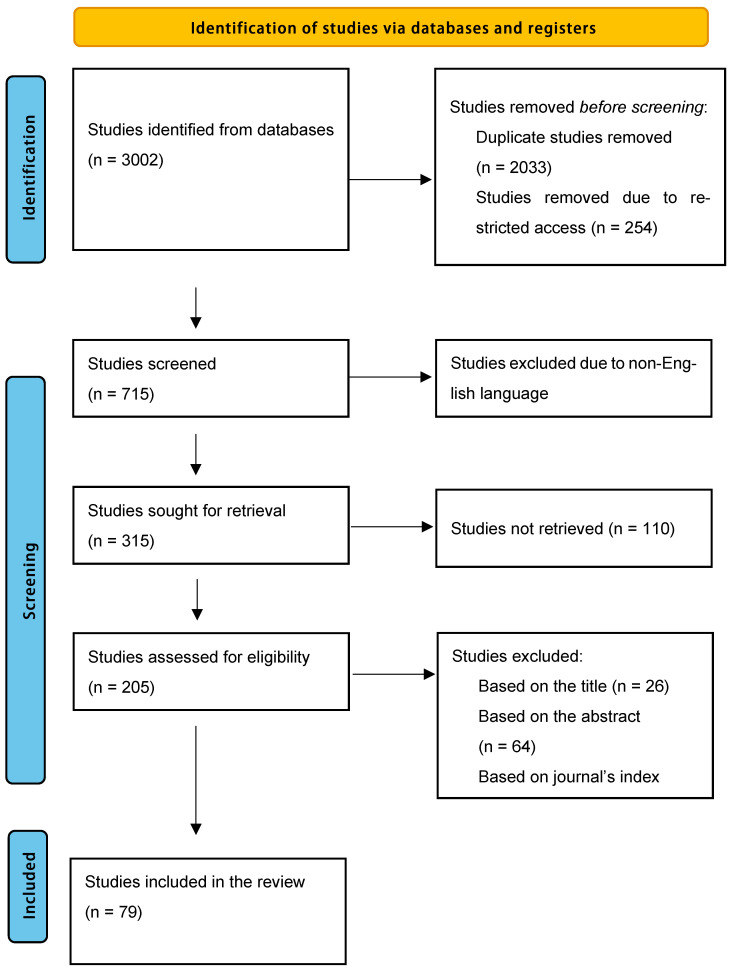
Identification of studies via databases and registers.

**Figure 2 healthcare-13-03157-f002:**
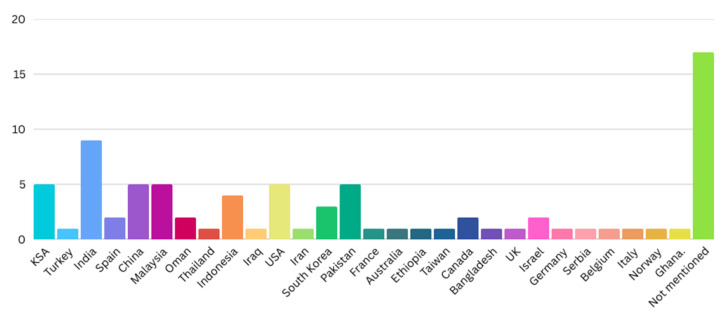
Global landscape of research on IoT adoption in healthcare.

**Figure 3 healthcare-13-03157-f003:**
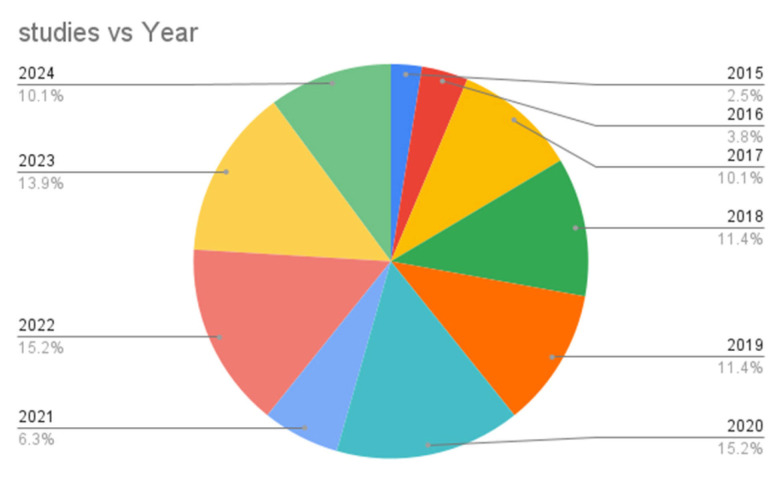
Distribution of selected publications across various years.

**Figure 4 healthcare-13-03157-f004:**
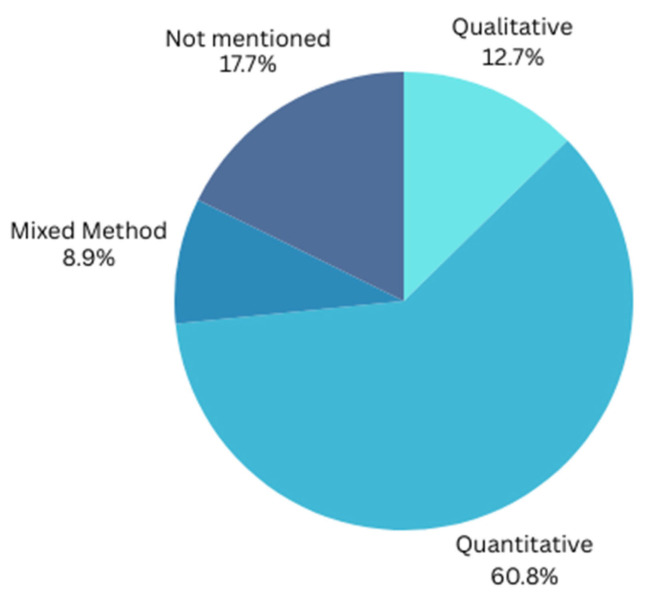
Research designs in reviewed studies on IoT acceptance in healthcare sector.

**Figure 5 healthcare-13-03157-f005:**
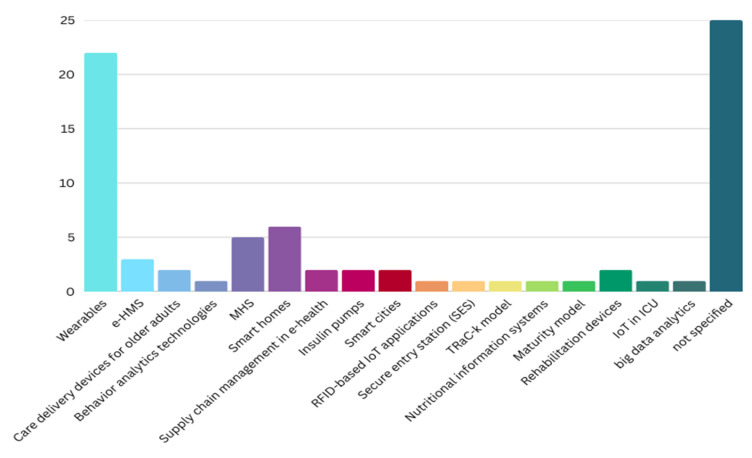
Technologies integrated with the Internet of Things (IoT).

**Figure 6 healthcare-13-03157-f006:**
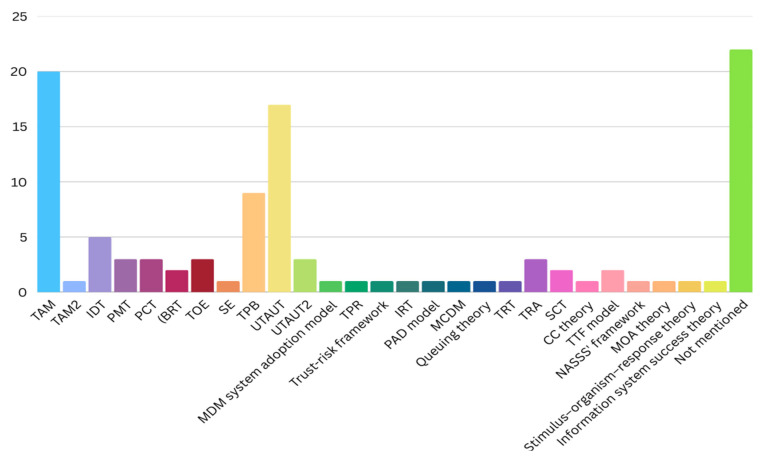
Predominant theoretical approaches in current research on IoT adoption in healthcare.

**Figure 7 healthcare-13-03157-f007:**
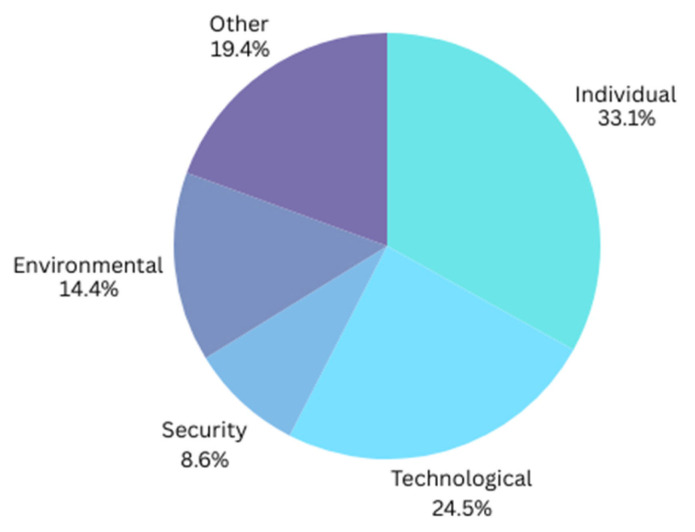
Research gaps per factor.

**Figure 8 healthcare-13-03157-f008:**
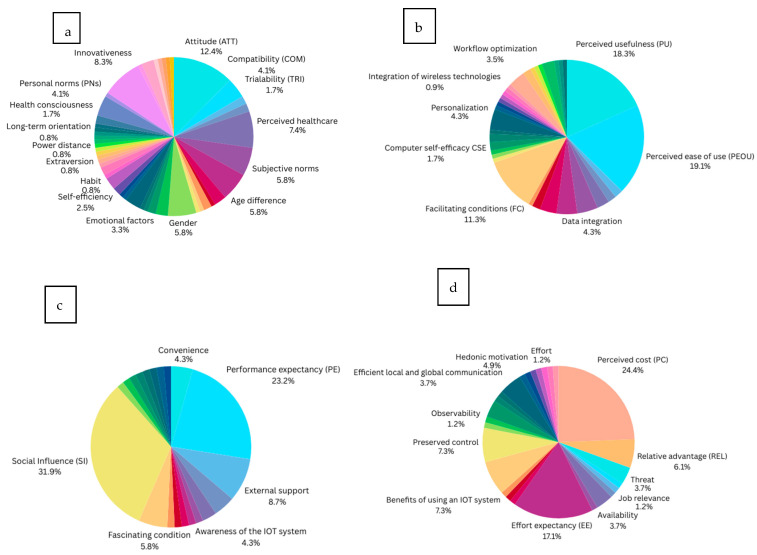
Individual factors influencing IoT adoption within the healthcare industry: (**a**) individual factors; (**b**) technological factors; (**c**) environmental factors; and (**d**) ‘other’ factors.

**Figure 9 healthcare-13-03157-f009:**
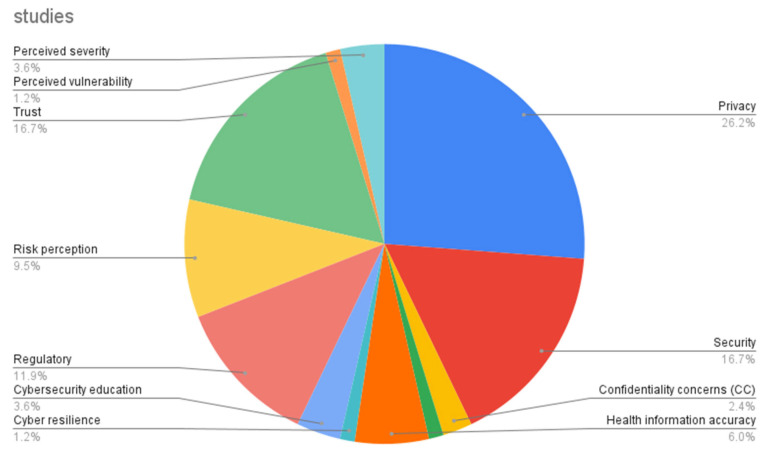
Research gaps in cybersecurity aspects.

**Table 1 healthcare-13-03157-t001:** Results and limitations of past studies on IoT adoption in healthcare.

Study	Sample Size	Period	Results	Limitations
[20]	146 articles	2015–2020	IoT research communities and engines were studied. Security-based categories were most frequently used (37%). Most articles aimed to enhance performance (22%), security (21%), and time (18%), with 57% of publications using simulation conditions.	Exclusive focus on articles published in respected journals. Use of 10 known online databases. Application of four questions. Classification of examined papers into five categories.
[21]	106 papers	2015–2022	Authors focused on three main aspects: motivation, challenges, and recommendations concerning the application of IoT to healthcare systems and service delivery. Most (77) were experimental studies. Most studies were published in IEEE Access; 22 were conducted in India, while 24 focused on remote health monitoring.	Articles within a specified time frame and with targeted goals were reviewed.
[22]	22 articles	2015–2021	The TAM, TPB, TRA, and UTAUT were the most commonly employed adoption theories. A total of 90 factors were identified: individual (16), technology (19), security (7), health (9), and environment (7). Preserved usefulness and ease of use were the most repeated adoption factors.	Focused only on physicians’ perspective on healthcare. Articles from nine databases that utilized quantitative methods only. Elements were categorized based on comprehensive, theoretical, and empirical research.
[23]	Not mentioned	Not mentioned	The authors categorized IoT applications in healthcare into monitoring, locating, and personalized medicine systems. Emphasized the importance of clear privacy policies, consent mechanisms, anonymization, and encryption techniques for successful adoption.	Ambiguity in details of the research process weakened the credibility and reliability of the results.
[8]	10 articles	Not mentioned	The study identified 94 factors influencing IoT adoption in healthcare, with these categorized into three primary themes: human behaviors, strategic decision-making, and technical enhancement.	Focused on peer-reviewed journal articles only. Potential bias in keyword selection.
Current research	79 articles	2015–2024	This research identified 28 theories or frameworks in the literature. A total of 28 countries/regions were involved, with India having the highest number of studies. Most studies (60.8%) utilized quantitative research methods. The years 2022 and 2019 each accounted for 15% of the selected publications. The research discovered 148 factors that were classified and analyzed into individual, technological, security, environmental, and ‘other’ categories.	

Notes: TAM = technology acceptance model; TPB = theory of planned behavior; TRA = theory of reasoned action; UTAUT = unified theory of acceptance and use of technology.

**Table 2 healthcare-13-03157-t002:** Eligibility criteria for study selection in the systematic literature review.

Criteria	
**Inclusions**	Studies focusing on factors that influence Internet of Things (IoT) adoption in healthcare; Published in a peer-reviewed journal with a high impact factor; English language; Available as full text; Published between 2015 and mid-2025.
**Exclusion**	Books, book chapters, and newspapers; Languages other than English; Not available as a full text; Published before 2015.

**Table 3 healthcare-13-03157-t003:** Number of studies per country.

Country	Studies
Kingdom of Saudi Arabia (KSA)	[31,32,33,34,35]
Turkey	[36]
India	[37,38,39,40,41,42,43,44,45]
Spain	[46,47]
China	[1,48,49,50,51]
Malaysia	[40,52,53,54,55]
Oman	[56,57]
Thailand	[40]
Indonesia	[40,58,59,60]
Iraq	[61]
USA	[62,63,64,65,66]
Iran	[67]
South Korea	[68,69,70]
Pakistan	[71,72,73,74,75]
France	[76]
Australia	[77]
Ethiopia	[78]
Taiwan	[79]
Canada	[80]
Bangladesh	[81]
United Kingdom (UK)	[82]
Israel	[83,84]
Germany	[85]
Serbia (Belgrade)	[86]
Belgium	[87]
Italy	[84]
Norway	[88]
Ghana	[89]
Not mentioned	[2,90,91,92,93,94,95,96,97,98,99,100,101,102,103,104,105]

**Table 4 healthcare-13-03157-t004:** Number of studies per year.

Year	Studies
2015	[46,48]
2016	[60,70,83]
2017	[36,41,62,68,85,93,99,102]
2018	[39,40,47,65,69,90,95,104,105]
2019	[31,35,42,43,57,59,80,98,106]
2020	[2,61,64,71,74,76,79,81,96,97,103,107]
2021	[54,56,72,91,94]
2022	[1,37,38,44,45,49,53,63,67,87,92,101]
2023	[32,34,52,55,58,66,73,77,78,82,88]
2024	[33,50,51,75,84,86,89,100]

**Table 5 healthcare-13-03157-t005:** Number of studies per research design.

Research Design	Studies
Qualitative	[35,45,65,67,80,82,85,87,88,107]
Quantitative	[1,2,31,32,34,36,37,38,40,41,42,44,46,47,49,50,52,53,54,55,56,57,58,59,60,61,63,64,66,68,69,70,72,73,74,75,76,77,78,79,81,86,90,92,97,98,103,106]
Mixed methods	[39,48,51,62,83,84,89]
Not mentioned	[33,43,71,91,93,94,95,96,99,100,101,102,104,105]

**Table 6 healthcare-13-03157-t006:** Technologies used in selected studies.

Focus Technology	Study
Personal IoT health devices, “wearables”	[1,35,36,39,44,48,50,60,66,69,70,73,74,77,78,79,85,92,97,103,106,107]
e-Health management system (e-HMS)	[37,49,104]
Care delivery devices for older adults	[65,105]
Behavior analytics technologies	[102]
Mobile health services (MHS)	[46,55,75,81,90]
Smart homes	[40,54,62,64,68,94]
Supply chain management in e-health	[34,45]
Insulin pumps	[88,93]
Smart cities	[41,99]
RFID-based IoT applications	[101]
Secure entry station (SES)	[58]
Telemedicine rounding and consulting for kids (TRaC-k) model	[80]
Nutritional information systems	[59]
Maturity model	[87]
Rehabilitation devices	[51,86]
IoT in intensive care units (ICUs)	[84]
IoT in big data analytics	[89]
Not specified	[2,31,32,33,35,38,42,43,47,52,53,55,56,57,61,67,71,72,76,82,83,91,95,96,100]

**Table 7 healthcare-13-03157-t007:** Theories used in selected studies.

Theory	Studies
Technology acceptance model (TAM), IoT acceptance model (IoTAM),	[1,31,36,41,46,49,51,58,60,61,63,68,70,77,84,86,90,106]
technological–personal–environmental (TPE) framework	[2,66]
Technology acceptance model 2 (TAM2), M2 competitive model	[46]
Innovation diffusion theory (IDT) (Diffusion of Innovation [DOI] theory),	[35,36,55,61]
consolidated framework for implementation research (CFIR)	[105]
Protection motivation theory (PMT)	[36,73,90]
Privacy calculus theory (PCT)	[36,56,103]
Behavioral reasoning theory (BRT)	[39,67]
Technology–organization–environment (TOE) model	[33,52,75]
Social exchange (SE) theory	[52]
Theory of planned behavior (TPB),	[41,55,56,58,69,86]
model of goal-directed behavior (MGB),	[44]
value–belief–norm (VBN) model	[50,55]
Unified theory of acceptance and use of technology (UTAUT),	[1,34,37,40,52,54,59,62,72,74,76,79,81,83,89]
unified theory of acceptance and use of technology–hospital staff (UTAUT-HS)	[32,89]
Unified theory of acceptance and use of technology 2 (UTAUT2)	[38,64,78]
Okazaki et al.’s (2015) mobile phone-based diabetes monitoring (MDM) system adoption model	[46]
Theory of perceived risk (TPR)	[1]
Trust–risk framework	[56]
Innovation resistance theory (IRT)	[53]
Pleasure, arousal, dominance (PAD) model	[62]
Multicriteria decision-making methods (MCDM) using analytic hierarchy process (AHP) and the technique for order preference by similarity to ideal solution (TOPSIS)	[71]
Queueing theory	[93]
Technology readiness theory (TRT)	[54]
Theory of reasoned action (TRA),	[41]
health belief model (HBM),	[72,106]
social cognitive theory (SCT), theoretical domains framework (TDF)	[69] [80]
Cybernetic control (CC) theory	[42]
Fit between the individuals, task, and technology (FITT) framework,	[105]
task–technology fit (TTF) model	[70]
Non-adoption, abandonment, scale-up, spread, sustainability (NASSS) framework.	[82]
Motivation, opportunity, ability (MOA) theory	[49]
Stimulus–organism–response theory	[51]
Information system success theory	[51]
Not mentioned	[43,45,47,48,57,65,77,85,87,88,91,92,94,96,97,98,99,100,101,102,107]

## Data Availability

The data presented in this study are available in publicly accessible databases, including the Research Information Sharing Service (RISS), CINAHL, MEDLINE, and PubMed. All data were derived from these public domain resources.

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
