# Peer review of "Exploring Factors Affecting the Adoption of IoT in Healthcare: A Systematic Literature Review"

_healthcare, 2025, doi:10.3390/healthcare13233157_

Round 1

Reviewer 1 Report

Comments and Suggestions for Authors

1. The paper identifies 139 unique indicators and consolidates them into five categories (individual, technological, security, environmental, other). However, the manuscript lacks a detailed description of the methodology used for this deduplication and classification. A rigorous protocol, such as thematic analysis with inter-coder reliability measures (e.g., Cohen's Kappa), is not mentioned. This omission raises concerns about the reproducibility and objectivity of the categorization, which is a core contribution of the review. The potential for researcher bias in assigning factors like "cost" to the 'other' category instead of a 'technological' or 'environmental' one is not addressed.

2. While cybersecurity is a stated focus, the analysis remains at a high, generic level (e.g., "privacy," "security"). A more technical review would map the identified security factors to the specific layers of a standard IoT architecture (Perception/Sensing, Network, Middleware, Application). For instance, "perceived vulnerability" could be linked to device-level attacks (e.g., node tampering), while "confidentiality concerns" relate to network and cloud data transmission. This layered analysis would provide a more actionable framework for addressing security in IoT healthcare systems.

3. The review compiles data from 79 studies, including the statistical significance (or lack thereof) of 139 factors across numerous contexts (Appendix A). This dataset presents a prime opportunity for a meta-analysis to quantify the effect sizes of the most frequently studied factors (e.g., Perceived Usefulness, Social Influence). The decision to forgo a meta-analysis in favor of a descriptive, frequency-based summary (e.g., Figure 7) limits the paper's ability to provide conclusive evidence on the strength and consistency of these relationships across the literature.

4. Given the finding that security factors constitute only 8.6% of the researched factors, what are the specific, technically-grounded causal mechanisms proposed by the reviewed literature that link specific security vulnerabilities (e.g., weak device authentication, unencrypted data at rest) to the behavioral intention of healthcare providers or patients to adopt IoT, as modeled by frameworks like UTAUT or TAM?

Author Response

  1. The paper identifies 139 unique indicators and consolidates them into five categories (individual, technological, security, environmental, other). However, the manuscript lacks a detailed description of the methodology used for this deduplication and classification. A rigorous protocol, such as thematic analysis with inter-coder reliability measures (e.g., Cohen's Kappa), is not mentioned. This omission raises concerns about the reproducibility and objectivity of the categorization, which is a core contribution of the review. The potential for researcher bias in assigning factors like "cost" to the 'other' category instead of a 'technological' or 'environmental' one is not addressed.

Agree, thank you for pointing this out, we have accordingly revised to emphasize this point

And we've modified to clarify that relying on Past Literature for Indicator Classification Is a Common and Accepted Practice. When classifying a large set of indicators, it is standard to rely on established frameworks and categorizations from prior literature to ensure consistency and comparability. Many systematic reviews and indicator taxonomies explicitly base their classification schemes on previous studies, established models, or widely accepted frameworks, rather than developing entirely new protocols for each review

This change can be found

Page number 14

Paragraph: 4.6. Factors influencing IoT adoption in healthcare sectors

Line 366-368

  1. While cybersecurity is a stated focus, the analysis remains at a high, generic level (e.g., "privacy," "security"). A more technical review would map the identified security factors to the specific layers of a standard IoT architecture (Perception/Sensing, Network, Middleware, Application). For instance, "perceived vulnerability" could be linked to device-level attacks (e.g., node tampering), while "confidentiality concerns" relate to network and cloud data transmission. This layered analysis would provide a more actionable framework for addressing security in IoT healthcare systems.

Agree, thank you for pointing this out, we have accordingly revised to emphasize this point

And we would like to clarify that while the current systematic literature review (SLR) acknowledges cybersecurity as a focal point, it intentionally adopts a broader perspective to explore the totality of factors influencing IoT adoption in healthcare. By examining a wide range of potential drivers and barriers, this study aims to lay the groundwork for future research, whether that pertains to technical security frameworks or other critical aspects of IoT implementation.

  1. The review compiles data from 79 studies, including the statistical significance (or lack thereof) of 139 factors across numerous contexts (Appendix A). This dataset presents a prime opportunity for a meta-analysis to quantify the effect sizes of the most frequently studied factors (e.g., Perceived Usefulness, Social Influence). The decision to forgo a meta-analysis in favor of a descriptive, frequency-based summary (e.g., Figure 7) limits the paper's ability to provide conclusive evidence on the strength and consistency of these relationships across the literature.

Agree, thank you for pointing this out, we have accordingly revised to emphasize this point

And we accordingly add a paragraph in the gaps section to clarify this matter

This change can be found

Page number 16

Paragraph: gaps and future agenda-IoT in the healthcare sector

Line: 464-469

  1. Given the finding that security factors constitute only 8.6% of the researched factors, what are the specific, technically-grounded causal mechanisms proposed by the reviewed literature that link specific security vulnerabilities (e.g., weak device authentication, unencrypted data at rest) to the behavioral intention of healthcare providers or patients to adopt IoT, as modeled by frameworks like UTAUT or TAM?

Agree, thank you for pointing this out, we have accordingly revised to emphasize this point

And we've found that we've mentioned only 2 scenarios of security factors regarding confidentiality concerns and the presence of clear objectives and plans in the (Security factors) paragraph on page 15, note that we have added more to better  link security vulnerabilities

This change can be found

Page number 15

Paragraph: security factors

Line: 409-420

Reviewer 2 Report

Comments and Suggestions for Authors

The manuscript presents a review of cybersecurity in the context of the Internet of Things (IoT) in healthcare. Given the exponential digitalization of healthcare systems and the rising cyber threat landscape, this topic holds significant academic, industrial, and policy relevance. The study’s effort to analyze 79 prior studies and classify cybersecurity determinants across multiple categories is commendable. However, the manuscript requires revision to improve conceptual clarity.

  1. The abstract should clearly indicate the methodology used (e.g., systematic literature review, bibliometric analysis, or scoping review). Mention of “comprehensive examination” is too vague.
  2. While the review covers a broad dataset (79 studies), its novelty is not distinctly articulated. The authors should clarify whether the novelty lies in the categorization framework, scope of synthesis, or geographical mapping of research efforts.
  3. A brief comparison with existing reviews (e.g., on IoT–healthcare security frameworks, blockchain integration, or AI-driven defense mechanisms) would help position the study within current literature.
  4. The introduction should better justify why this review is needed now, perhaps by referencing recent cybersecurity incidents or rapid IoT expansion in post-pandemic healthcare.
  5. The reference to “technologies and theories/frameworks” is valuable but underexplored. The manuscript should highlight which frameworks (e.g., Technology Acceptance Model, Protection Motivation Theory, or Risk Perception Theory) were most frequently used.
  6. Include a concise explanation of how the categorization contributes to theory development in healthcare cybersecurity, does it extend existing models or propose a new integrated conceptual framework?
  7. Discuss how environmental or individual factors (such as digital literacy or organizational culture) interact with technological determinants to influence vulnerability or resilience.
  8. The discussion should link results to policy implications, for example, recommendations for healthcare institutions, regulators, or technology developers.
  9. The conclusion should move beyond general statements (“strengthen security measures”) toward actionable outcomes such as developing standardized IoT cybersecurity frameworks, cross-border data governance models, or AI-driven threat prediction systems. Include potential research directions: e.g., integrating edge computing, blockchain-based verification, or privacy-preserving federated learning into IoT healthcare security. Discuss ethical and legal dimensions—especially compliance with HIPAA, GDPR, and emerging AI Act frameworks.
  10. The introduction should supplement the current research and enrich the content of the paper. For example: https://doi.org/10.3389/fpubh.2025.1635475, https://doi.org/10.3390/jtaer20030231, https://doi.org/10.1109/JIOT.2025.3525623

Author Response

  1. The abstract should clearly indicate the methodology used (e.g., systematic literature review, bibliometric analysis, or scoping review). Mention of “comprehensive examination” is too vague.

Agree, thank you for pointing this out, the abstract clearly indicates that this paper employs a systematic literature review (SLR) to investigate the factors influencing IoT adoption in healthcare, clarify research trends, and examine the impact of cybersecurity

This can be found in the method section of the abstract

Page number 1

Line 13-15

  1. While the review covers a broad dataset (79 studies), its novelty is not distinctly articulated. The authors should clarify whether the novelty lies in the categorization framework, scope of synthesis, or geographical mapping of research efforts.

Agree, thank you for pointing this out. This study distinguishes itself by integrating several examinations into a singular analysis while investigating the used theories, the countries of the studies, the publication year, and the research methodology. The majority of existing current research encompasses two or fewer of these topics. It also emphasizes the importance of implementing more theories to better understand the factors finally, it draws attention to the need for more deeply focused security-related research, with the findings providing guidelines for future research focused on IoT adoption within the healthcare sector.

This explanation can be found

Page number:19- 20

Paragraph: contribution (research perspective), conclusion

Line: 619-625 , 631-633

  1. A brief comparison with existing reviews (e.g., on IoT–healthcare security frameworks, blockchain integration, or AI-driven defense mechanisms) would help position the study within current literature.

Agree, thank you for pointing this out. In comparison to existing reviews that focus on technical aspects such as IoT-healthcare security frameworks, blockchain integration, and AI-driven defense mechanisms, my study distinctly centers on the adoption factors of IoT in the healthcare sector. While those studies emphasize the technological solutions to security and operational efficiency, my research aims to explore the socio-economic and organizational barriers and facilitators influencing the integration of IoT technologies within healthcare practices. To further delineate this focus, I have included a table summarizing the results and limitations of past studies on IoT adoption in healthcare, which highlights the gaps in current literature. This contextual positioning not only underscores the unique contributions of my study but also demonstrates its relevance in addressing the critical need for understanding the human and organizational elements necessary for successful IoT adoption in healthcare settings. By concentrating on adoption aspects, my research aims to provide actionable insights that can drive effective implementation and utilization of IoT technologies, ultimately enhancing patient care and operational efficiency.

Page number 3- 4

Paragraph Table 1

  1. The introduction should better justify why this review is needed now, perhaps by referencing recent cybersecurity incidents or rapid IoT expansion in post-pandemic healthcare.

Agree, thank you for pointing this out. We have accordingly revised to emphasize this point

And we modified the script by adding the justification

This change can be found

Page number:2

Paragraph: introduction

Line:87-93

  1. The reference to “technologies and theories/frameworks” is valuable but underexplored. The manuscript should highlight which frameworks (e.g., Technology Acceptance Model, Protection Motivation Theory, or Risk Perception Theory) were most frequently used.

Agree, thank you for pointing this out, In the results section, abstract, and conclusion, I have highlighted that the Technology Acceptance Model (TAM) and its variants emerged as the dominant frameworks in the literature. This focus underscores the importance of user acceptance in understanding technology adoption.

This can be found results section, the abstract, and the conclusion

Page number 1- 12-18

  1. Include a concise explanation of how the categorization contributes to theory development in healthcare cybersecurity, does it extend existing models or propose a new integrated conceptual framework?

Agree, thank you for pointing this out, we have accordingly revised to emphasize this point

And we updated the conclusion to better explain how the categorization contributes to theory development in healthcare cybersecurity.

This change can be found

Page number 20

Paragraph: conclusion

Line: 637-642

  1. Discuss how environmental or individual factors (such as digital literacy or organizational culture) interact with technological determinants to influence vulnerability or resilience.

Agree, thank you for pointing this out. We have accordingly revised to emphasize this point

And we've accordingly updated the Factors section to help clarify how environmental or individual factors interact with technological determinants to influence vulnerability or resilience.

This change can be found

Page number 17

Paragraph : Factors

Line : 524-532

  1. The discussion should link results to policy implications, for example, recommendations for healthcare institutions, regulators, or technology developers.

Agree, thank you for pointing this out, we have accordingly revised to emphasize this point

And we've added a new subsection called Future recommendations  with all potential recommendations

This change can be found

Page number:19

Paragraph: Future recommendations

Line: 593-603

  1. The conclusion should move beyond general statements (“strengthen security measures”) toward actionable outcomes such as developing standardized IoT cybersecurity frameworks, cross-border data governance models, or AI-driven threat prediction systems. Include potential research directions: e.g., integrating edge computing, blockchain-based verification, or privacy-preserving federated learning into IoT healthcare security. Discuss ethical and legal dimensions—especially compliance with HIPAA, GDPR, and emerging AI Act frameworks.

Agree, thank you for pointing this out, we have accordingly revised to emphasize this point

And we updated the conclusion accordingly

This change can be found

Page number: 20

Paragraph: conclusion

Line: 663-681

  1. The introduction should supplement the current research and enrich the content of the paper. For example: https://doi.org/10.3389/fpubh.2025.1635475, https://doi.org/10.3390/jtaer20030231, https://doi.org/10.1109/JIOT.2025.3525623

Agree, thank you for pointing this out, we have accordingly revised to emphasize this point

And we’ve updated the introduction accordingly

This change can be found

Page number 2

Paragraph: introduction

Line:55-63

Reviewer 3 Report

Comments and Suggestions for Authors

The authors in this paper conduct a literature review to compile and analyze different factors that influence the adoption of IoT in the field of healthcare. The main points of focus are the factors affecting the implementation of IoT, academic trends, and understanding the role and impact of cybersecurity within the field of Healthcare.

However:

1- A big part of the paper is on how the authors identified studies for their work

2- Some of the key attributes used in section 4 are high-level analytical attributes (e.g., the number of studies in each country, the number of publications in each year) without more details on the reasons/meaning behind these numbers.

3- core sections like 4.3, 4.4, and 4.5,  the authors listed the studies where the corresponding technology or research design approach was utilized without providing more details about such technologies or design approaches.

1- 

Author Response

1- A big part of the paper is on how the authors identified studies for their work

Agree, thank you for your feedback. We recognize that the study selection process is a critical component of our research methodology. In our paper, we implemented a systematic approach to identify relevant literature, utilizing well-defined criteria for inclusion and exclusion that reflect the specific focus on IoT adoption factors. By doing so, we aimed to ensure a comprehensive overview of the current landscape and to capture a diverse range of perspectives on this important topic.

2- Some of the key attributes used in section 4 are high-level analytical attributes (e.g., the number of studies in each country, the number of publications in each year) without more details on the reasons/meaning behind these numbers.

Agree, thank you for pointing this out, we have accordingly revised to emphasize this point

And we’ve mentioned the importance of mentioning these numbers lies in their ability to reveal gaps in research, whether related to specific countries, underutilized theories, or the predominance of certain research methods, such as quantitative versus qualitative approaches. By highlighting these gaps, the data not only provides a clearer picture of the current state of IoT adoption in healthcare but also paves the way for future research in these areas. This insight encourages researchers to explore underrepresented regions, theories, and methodologies, ultimately contributing to a more balanced and comprehensive understanding of the field.

This can be found

Page number: 20

Paragraph: Conclusion

Line: 656- 662

3- core sections like 4.3, 4.4, and 4.5,  the authors listed the studies where the corresponding technology or research design approach was utilized without providing more details about such technologies or design approaches.

Agree, thank you for pointing this out, we would like to clarify that these sections provide a detailed examination of the research design approaches, technologies used, and theories employed in the 79 studies investigated. Each section includes comprehensive tables that delineate the various methodologies and interconnected technologies or theories utilized to explore the factors influencing IoT adoption in healthcare. This structured presentation aims to offer a clear understanding of the studies and their respective approaches

This change can be found in the findings / Results chapter

Page number: starting from page 7

Round 2

Reviewer 1 Report

Comments and Suggestions for Authors

Based on my review of the revised manuscript and the authors' responses to previous comments, I can confirm that the authors have diligently addressed the reviewers' feedback and significantly improved the quality of the manuscript. The revisions are well-justified, scientifically sound, and enhance the clarity and depth of the study.

However, to further strengthen the manuscript and align it with emerging research trends, I recommend that the authors consider discussing or incorporating the following cutting-edge concepts and technologies in their future research agenda or discussion sections. These elements are highly relevant to IoT in healthcare and can provide a more comprehensive outlook:

Suggested Topics for Future Research & Discussion

  1. Load-balanced and QoS-aware Software-Defined Internet of Things (SD-IoT)

    • With the increasing number of connected devices in healthcare, network congestion and resource allocation become critical.

    • Future studies could explore how Software-Defined Networking (SDN) can be used to dynamically manage network resources, ensure Quality of Service (QoS), and support real-time health monitoring systems.

  2. Software-Defined Internet of Multimedia Things (SD-IoMT)

    • Multimedia data (e.g., medical imaging, video consultations) is becoming integral to telehealth.

    • Research is needed on how SDN-enabled IoMT can efficiently handle high-bandwidth multimedia streams while ensuring low latency and security.

  3. Multimedia Streaming in Software-Defined Internet of Vehicles (SD-IoV) for Healthcare

    • In mobile healthcare scenarios (e.g., ambulances, patient transport), reliable multimedia streaming is essential.

    • Future work could investigate how SD-IoV can support real-time video consultations and data transmission between vehicles and hospitals.

  4. Task Scheduling in Fog-IoT Networks Using Hybrid Algorithms (e.g., Aquila Optimizer and Whale Optimization Algorithm)

    • Fog computing brings computational resources closer to IoT devices, reducing latency.

    • Efficient task scheduling is crucial. The combination of Aquila Optimizer (AO) and Whale Optimization Algorithm (WOA) could be explored to optimize resource usage and response time in healthcare Fog-IoT environments.

Author Response

Thank you for your insightful comments and suggestions regarding our manuscript. We appreciate the time you took to review our work.

We have decided to incorporate one of your future suggestions ( Multimedia Streaming in Software-Defined Internet of Vehicles (SD-IoV) for Healthcare) , as we believe it is the most relevant to our study. This point has been emphasized in the literature we have examined, further supporting its inclusion in our work.

this change can be found in 

5.4. Future recommendations

line: 605-609

Reviewer 3 Report

Comments and Suggestions for Authors

The authors addressed the different questions/comments mentioned regarding the previous version.

Author Response

Thank you for your valuable feedback on our manuscript. We want to let you know that we have already improved the figures and redesigned the tables to enhance clarity and presentation.